# Early life experience sets hard limits on motor learning as evidenced from artificial arm use

Roni O Maimon-Mor[1,2]*, Hunter R Schone[2,3], David Henderson Slater[4], A Aldo Faisal[5], Tamar R Makin[2]

[1]WIN Centre, Nuffield Department of Clinical Neuroscience, University of Oxford, Oxford, United Kingdom; [2]Institute of Cognitive Neuroscience, University College London, London, United Kingdom; [3]Laboratory of Brain & Cognition, NIMH, National Institutes of Health, Bethesda, United States; [4]Oxford Centre for Enablement, Nuffield Orthopaedic Centre, Oxford, United Kingdom; [5]Departments of Bioengineering and of Computing, Imperial College London, London, United Kingdom

**Abstract** The study of artificial arms provides a unique opportunity to address long-standing questions on sensorimotor plasticity and development. Learning to use an artificial arm arguably depends on fundamental building blocks of body representation and would therefore be impacted by early life experience. We tested artificial arm motor-control in two adult populations with upper-limb deficiencies: a congenital group—individuals who were born with a partial arm, and an acquired group—who lost their arm following amputation in adulthood. Brain plasticity research teaches us that the earlier we train to acquire new skills (or use a new technology) the better we benefit from this practice as adults. Instead, we found that although the congenital group started using an artificial arm as toddlers, they produced increased error noise and directional errors when reaching to visual targets, relative to the acquired group who performed similarly to controls. However, the earlier an individual with a congenital limb difference was fitted with an artificial arm, the better their motor control was. Since we found no group differences when reaching without visual feedback, we suggest that the ability to perform efficient visual-based corrective movements is highly dependent on either biological or artificial arm experience at a very young age. Subsequently, opportunities for sensorimotor plasticity become more limited.

*For correspondence:
r.maimon@ucl.ac.uk

## Introduction

We move our hands with such apparent ease, yet the underlying process involves complex computations, representations, and integration of information across multiple systems and modalities (*Scott, 2004*; *Wolpert, 1997*). Learning to move our limbs precisely and accurately begins in utero, where embryos have been documented refining arm-to-mouth reaching movements (*Zoia et al., 2007*). The trajectory of optimizing reaching across infancy (*Berthier and Keen, 2006*; *Leed et al., 2019*) and childhood (*Contreras-Vidal, 2006*; *Schneiberg et al., 2002*; *Simon-Martinez et al., 2018*; *Sveistrup et al., 2008*) is highly protracted, roughly plateauing at around 10–12 years of age. In the present study, we investigated reaching behavior in two groups of individuals who experienced a vastly different motor development but share current motor constraints: individuals with a congenital unilateral upper-limb difference (i.e., born with a partial upper-limb, missing a hand and a part of their arm; hereafter congenital group) and individuals who were born with a fully developed upper-limb but lost it as adults following amputation (hereafter acquired group). We asked how a sensorimotor system

that developed with (acquired group) or without (congenital group) experience of a complete arm supports the control of an upper-limb substitute (artificial arm). Artificial arm motor control provides a unique opportunity to address key questions surrounding sensorimotor plasticity. The flexibility needed to support this new body part is arguably different from that observed in traditional motor learning paradigms (e.g., involving tools) as it might relate to more fundamental building blocks of body representation and the internal models for motor control.

We consider three possible predictions, involving differences in artificial arm motor control across these two groups: first, perhaps the most straightforward prediction is that the congenital group's artificial arm motor control would be superior to that of the acquired group. It is often thought that the brain is more plastic during earlier stages of development (*Knudsen, 2004*). Therefore, it becomes more difficult to acquire radically new motor skills in adulthood, which is probably why most virtuoso musicians and athletes started practicing their trade in their childhood (*Penhune, 2011*). As mentioned above, individuals with a congenital limb difference start using artificial arms at a very young age (in our sample as early as 3 months with an average of ~2.5 years), even before early training for musical and athletic skills. It therefore stands to reason that in comparison to the acquired group, who only begin to learn to use their artificial arm as adults (in our sample at a mean age of 32), the congenital group should have had more time and practice in early childhood to perfect their artificial arm motor skill. Moreover, individuals with an acquired limb difference often experience a 'phantom hand' (*Stankevicius et al., 2021*), rooted in a maintained representation of their missing arm (*Bruurmijn et al., 2017*; *Kikkert et al., 2016*; *Wesselink et al., 2019*) which might in theory interfere with the acquisition of a representation of an arm substitute (the artificial arm). Perhaps most importantly, relative to individuals with an acquired limb difference, individuals with a congenital limb difference tend to make better use of their artificial arm in daily life (*Biddiss and Chau, 2007*). Taken together, these considerations lead to a strong hypothesis that the congenital group would have had better opportunities for developing sensorimotor artificial arm control.

A second alternative hypothesis is that early life disability, as experienced by the congenital group, might offset motor development, such that their disability-related impairment would not necessarily lead to inferior motor performance with an artificial arm. In other words, that the performance in the congenital group would be equivalent to that of the acquired group. It could be argued that regardless of the undisputed role of early life experience in shaping brain organization and function, the canonical brain infrastructure will still exist and be able to support the dormant function, even in the congenital group. For example, in the visual domain, children born with high-density cataracts who received corrective surgery later in life have been shown to retain some rudimentary forms of visual perception (*Gandhi et al., 2017*). This hypothesis is consistent with recent studies emphasizing normal visuomotor processes and representations of hands of individuals born with no hands (*Vannuscorps and Caramazza, 2015*; *Vannuscorps and Caramazza, 2016* though see *Maimon-Mor et al., 2020b*; *Philip et al., 2015*; *Philip and Frey, 2011*; *Wesselink et al., 2019*). Moreover, considering these individuals potentially have a lifetime of daily experience controlling an artificial arm, it is possible that they will be able to 'close the gap' that had started in early development, relative to their able-bodied peers. Indeed, it has been consistently shown how well adults can adapt their motor behavior to overcome a myriad of perturbations, and learn to perform intricate and skillful tasks (*Wolpert et al., 2011*).

A third hypothesis asserts that experience with a complete arm early in life might be crucial for the successful integration of any arm, including an artificial one. Therefore, motor control of an artificial arm would be superior in the acquired group who had 'typical' motor development for their missing arm, relative to the congenital group who had atypical motor development. This idea is rooted in the old debate of the relative contributions of nature versus nurture. Current views consider neural development an interaction between predetermined maturation based on a genetic template and experience (*Adolph and Franchak, 2017*; *Karmiloff-Smith, 1998*; *Krubitzer and Prescott, 2018*). The neural topographical organization of sensory input across the cortex has been shown to be in part determined by genetics (*Miyashita-Lin et al., 1999*; *Rubenstein et al., 1999*). However, for both the motor system and the visual system (an integral input to the sensorimotor loop), early deprivation has been shown to have a permanent effect on development (*de Heering et al., 2016*; *Walton et al., 1992*). As such, individuals who, prior to their amputation, benefited from a typical developmental trajectory might be able to rely on the existing upper-limb infrastructure, after amputation, when learning to control an artificial arm. This is in stark contrast to the congenital group, who never

developed an upper limb, due to a developmental malformation, and therefore lack both visual and motor experience of their missing limb during the formative years of their motor development. Based on this hypothesis, the acquired group would have superior motor control of artificial arms, compared to the congenital group.

Consistent with this final hypothesis, we found that although they had started training to use an artificial arm earlier in life and sustained more elapsed years of artificial arm use, individuals with a congenital limb difference were unable to refine their reaching control to normal levels. The congenital group produced larger reaching errors with their artificial arms compared to both artificial arm reaches of amputees and nondominant hand reaches of age-matched controls. We used numerous measures and tasks to interrogate the potential contributors to sensorimotor performance across groups, allowing us to disentangle the different components that might have contributed to the afore-mentioned group differences. Finally, we explored how the key components contributing to reduced motor control relate to early life experience with an artificial arm. Our results suggest that the forma-tion of an arm representation in early life has a long-lasting effect on the incorporation of an artificial arm, highlighting that opportunities for sensorimotor plasticity become more limited with age, even across early childhood.

## Results

### The congenital group shows inferior artificial arm motor control

In order to assess artificial arm motor control, participants performed visually guided reaches to a set of targets using a robotic manipulandum device (see *Figure 1A and B*). Motor control measures were compared across three groups: a congenital group (n=18), an acquired group (n=14), and an age-matched control group (n=19). All included participants were able to control the robotic handle and perform the task using the same speed-accuracy trade-off parameters following Fitts' Law (see Appendix 1). Reaching performance was evaluated by measuring the mean absolute error participants made across all targets (see *Figure 1C*). The absolute error refers to the distance from the cursor's position at the end of the first reach (endpoint) to the center of the target in each trial. The endpoint of each trial was set as the arm location at the end of the first reach, identified using the trial's kine-matic data (see Materials and methods). Participants completed the same task using their intact arm as well, allowing us to control for individual differences relating to aspects of the task that are not artificial arm-specific. We found no significant group effect ($F_{(2,48)}=1.05$, p=−0.14) when comparing the absolute errors of the intact arm and dominant arm across the three groups (control, acquired, and congenital groups). However, this result was inconclusive ($BF_{10}=0.65$), that is, supports neither the null nor the alternative hypothesis. We therefore included the intact arm as a confound regressor in subsequent analyses (see Early life but not present experience with artificial arms affects adult motor control in individuals with a congenital limb difference for additional results and discussion regarding intact arm performance).

We performed an analysis of covariance (ANCOVA) on participants' artificial arm errors where participants' intact arm errors were defined as a covariate, and group as a between-subjects vari-able. We found a significant effect of intact-hand performance ($F_{(1,47)}=28.65$, p<0.001, $\eta_p^2=0.38$), that is, participants who had small errors with their intact arm also tended to have smaller errors with their nondominant/artificial arm. We found a significant group effect ($F_{(2,47)}=13.81$, p<0.001, $\eta_p^2=0.37$), indicating the groups differed in their visuomotor performance with their artificial arm (or nondominant arm in controls). Post hoc comparisons revealed that the congenital group exhib-ited larger errors with their artificial arm compared to both the artificial arm of the acquired group (t=−3.77, $p_{tukey}=0.001$, Cohen's-d=−1.39), and the nondominant arm of the control group (t=−5.06, $p_{tukey}$ <0.001, Cohen's-d=−1.705). Conversely, the acquired group's artificial arm errors did not differ from those of the control group's nondominant arm (t=−0.885, $p_{tukey}=0.65$), indicating a specific deficit in error reaching for the congenital groups' artificial arm. To further explore the non-significant performance difference between the acquired and control groups, we used a Bayesian approach (*Rouder et al., 2009*), that allows for testing of similarities between groups (the null hypothesis). In this analysis, the smaller effect size of the two reported here (1.39) was inputted as the Cauchy prior width. The resulting Bayesian Factor ($BF_{10}=0.28$) provided moderate support to the null hypothesis (i.e., smaller than 0.33). To summarize, the congenital group shows significantly inferior artificial arm

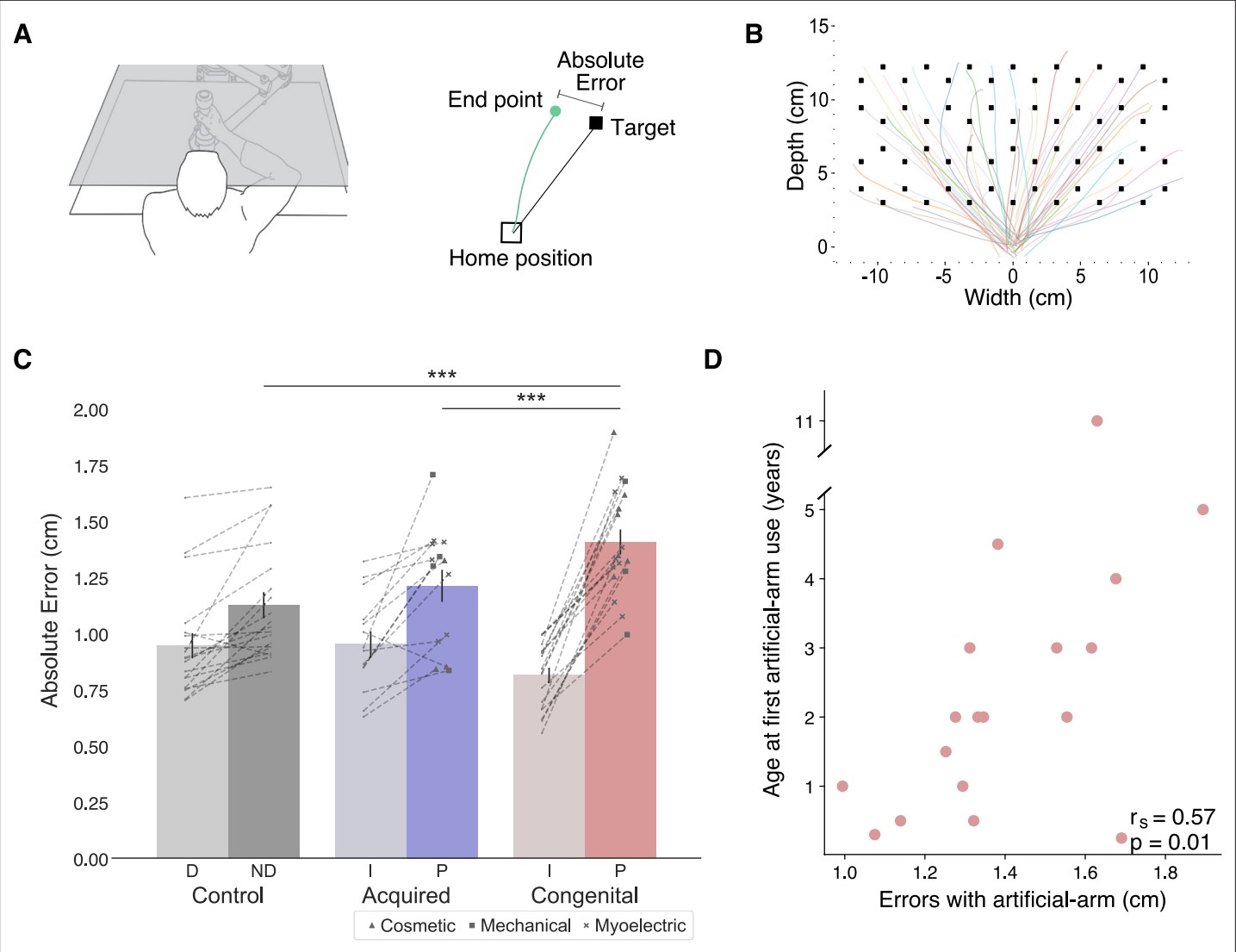

**Figure 1.** Experimental design and main analyses. (**A**) Left: An illustration of the robotic manipulandum device setup. Participants performed reaching movements while holding a robotic handle. A monitor displaying the task components was viewed via a mirror, such that participants did not have direct vision of their arm. Visual feedback was provided as a cursor depicting the current location of the arm. Right: A visualization of a single trial and the different terms used. In each trial, participants reached from the home position to a single visual target. The green line represents the participant's arm trajectory. (**B**) Reaching trajectories to all targets from a randomly selected participant. The different colored lines are trajectories of individual reaching trials. (**C**) Reaching performance as measured by absolute errors for each group for each arm. Gray, blue, and red colors represent control, acquired, and congenital groups, respectively. Lighter colors represent intact/dominant-arm performance; darker colors represent artificial/nondominant-arms. We found a significant group effect ($F_{(2,47)}$=13.81, p≤0.001, $\eta_p^2$=0.37), with the congenital group making larger errors with their artificial arm compared to both acquired group's artificial arm (t=−3.77, $p_{tukey}$=0.001, Cohen's-d=−1.39) and control group's nondominant arm (t=−5.06, $p_{tukey}$ <0.001, Cohen's-d=−1.705). Dotted lines connect errors between arms of individual participants. Artificial arm markers represent artificial arm types. (**D**) Relationship between age at first artificial arm use and artificial arm reaching errors in the congenital group. D – Dominant arm, ND – Nondominant arm, I – Intact arm, A – Artificial arm. ***p<0.001.

The online version of this article includes the following figure supplement(s) for figure 1:

**Figure supplement 1.** Intact hand errors and daily artificial arm use.

reaching accuracy in our task compared to the other groups (see *Supplementary file 1a* for results of the statistical analyses confirming that this effect was not driven by the side (L/R) of the artificial arm/ nondominant side).

## Physical aspects of artificial arm use do not correlate with endpoint errors

We first wanted to rule out the influence of two crucial physical aspects of artificial limb use: residual limb length and device type. The length of the residual limb, used to carry and control the artificial arm, can have a potential impact on the level of its motor control. The shorter the residual limb, the more restrictive the artificial limb control is, for example, due to more restrictive motion and less leverage. Across both acquired and congenital groups, the correlation between absolute reaching error and either residual limb length was not significant ($r_s(30)=-0.23$, $p=0.2$). Moreover, repeating the previously reported ANCOVA analysis while adding residual limb length as a covariate revealed no significant effect of residual limb length ($p>0.2$) and, importantly, did not abolish the group effect ($p=0.03$; for a full statistical report see *Supplementary file 1b*). Therefore, the length of the residual limb does not play a significant role in the observed group effect for endpoint accuracy of artificial arm reaches.

It is important to note that while artificial arm devices have different levels of wrist- and grasp-control, they are all used similarly during reaching (i.e., in the current task the participants did not use any of the devices additional control features). Yet, we wished to confirm that differences in artificial arm types used across the two groups did not affect our findings. The devices used by our participants can generally be categorized into three device types: (1) 'cosmetic' devices that look like a hand (n=10), these are static devices that do not afford additional control, (2) 'body-powered' devices in the shape of a hook (n=7), these include a mechanical grip control, and (3) 'myoelectric' devices (n=15), these are relatively heavy devices, controlled using signals from the muscles of the residual limb and powered by motors to perform grip functions (see marker type in *Figure 1C*). Despite the differences in appearance and control mechanisms between devices, the type of device used does not seem to influence endpoint reaching error in our task, as demonstrated by a one-way ANOVA with device type as a between-subject variable showing no significant differences between devices ($F_{(2,29)}=0.435$, $p=0.65$, $BF_{10}=0.275$).

## Artificial arm errors in the congenital group originate from increased motor noise

In our analyses so far, we reported the absolute error—the average distance of the endpoint from the visual target across all reaches. An increase in absolute error can be the result of two different type of error components (see *Figure 2A*): bias (e.g., consistently reaching to the left of the target) and noise (variability/spread of endpoints). These are often also referred to as accuracy and precision, respectively. A larger bias is caused by a model-mismatch, for example, an inaccurate internal forward model that produces a biased control policy that consistently fails to accurately transport the arm to the correct location, resulting in poor accuracy. Several different sources can cause a noisier performance, for example, large uncertainties in the sensory estimates of proprioception (*Gordon et al., 1995*), motor noise (*Faisal et al., 2008*), or a result of a failed computation (*Contreras-Vidal, 2006*), to optimally use sensory inputs to reduce this inherent noise. Assessing these error components separately can give us an insight into the underlying processes that are affected in the congenital group.

In order to calculate these measures across all targets, we drew the error vector for each trial (the line connecting the target with the endpoint location) and overlaid all the error vectors, as if they were made to a single target. While error vectors are known to be location-dependent (*Van Beers et al., 2004*; *Van Beers et al., 1999*), because we compare bias and noise measures across groups, and the distribution of error vector directions did not differ between groups (Watson-Williams circular test: $F_{(2,48)}=1.95$, $p=0.15$; see *Figure 2A*), these measures are suitable for present purposes. We compared artificial arm and nondominant arm biases (distance from the center of the endpoint to the target) across groups, using intact arm biases as a covariate. The ANCOVA resulted in no significant (inconclusive) group differences ($F_{(2,47)}=2.40$, $p=0.1$, $BF_{Incl}=0.72$; see *Figure 2A*). When comparing artificial arm and nondominant arm motor noise (spatial standard deviation [SD] of endpoints relative to the center of the endpoints), using intact-hand noise as a covariate, we found a significant group

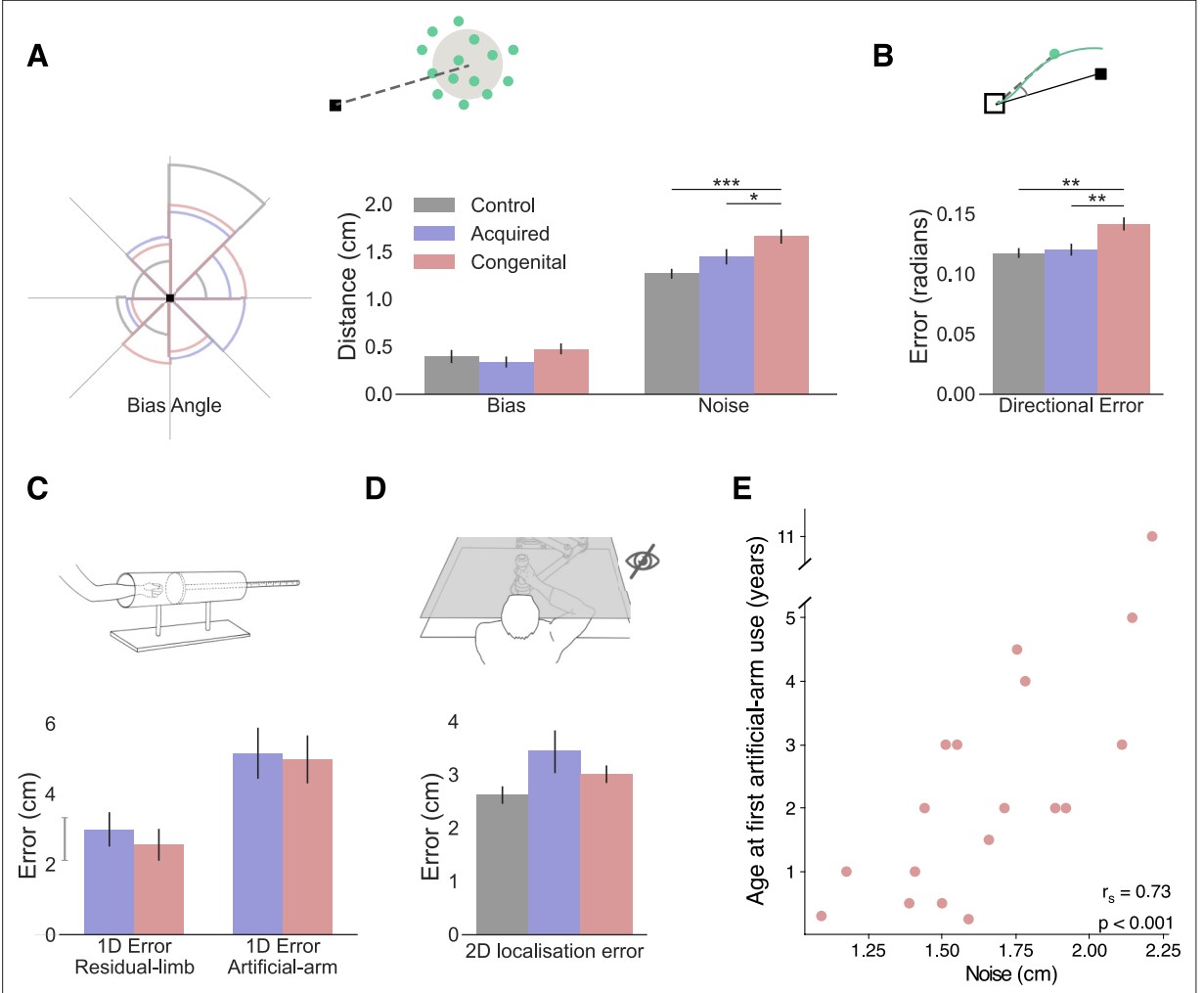

**Figure 2.** Exploring the source of increased reaching errors using additional analyses and tasks. In all plots, gray, blue, and red represent the control, acquired, and congenital groups, respectively. (**A**) Left: rose plot density histogram of the distribution of bias angles across the groups, the larger the arc the more individuals from that groups had a bias within the arcs angle range. We found no significant differences in bias angle between the groups (Watson-Williams circular test: $F_{(2,48)}$=1.95, p=0.15). Right: Error bias and noise results. No significant group differences were found for bias ($F_{(2,47)}$=2.40, p=0.1, $BF_{Incl}$=0.72). The congenital group shows significantly more motor noise than amputees and controls ($F_{(2,47)}$=14.15, p<0.001, $\eta_p^2$=0.38; post hoc significance levels are plotted). (**B**) Initial directional error results. The congenital group has larger directional error in the initial phase of reaching ($F_{(2,47)}$=8.01, p<0.001, $\eta_p^2$=0.26; post hoc significance levels are plotted). (**C**) 1D localization task results. Participants placed their residual limb or artificial arm inside an opaque tube and were asked to assess the location of the limb using their intact arm. We found no localization differences between the acquired and congenital groups in either condition ($BF_{10}$ <0.33 for both). The gray line next to the y-axis shows the mean ± s.e.m of control group's nondominant hand localization errors. (**D**) 2D localization task results. Using the same apparatus, participants performed reaches to visual targets without receiving visual feedback during the reach. We found no group differences in absolute error ($F_{(2,44)}$=0.71, p=0.5, $BF_{Incl}$=0.33). (**E**) Relationship between artificial arm motor noise and age at first artificial arm use artificial arm in the congenital group. See *Figure 2—figure supplement 1* for plots with individual participants' data points. *p<0.05, **p<0.01, ***p<0.001.

The online version of this article includes the following figure supplement(s) for figure 2:

**Figure supplement 1.** Plots with individual participants' data points.

effect ($F_{(2,47)}$=13.205, p<0.001, $\eta_p^2$=0.36; see *Figure 2A*). Reflecting the absolute error findings, the congenital group exhibited larger motor noise with their artificial arm compared to the artificial arm of the acquired group (t=−2.90, $p_{tukey}$=0.015, Cohen's-d=−0.855), and the nondominant arm of control group (t=−5.31, $p_{tukey}$ <0.001, Cohen's-d=−1.65). Comparing motor noise between amputees' artificial arm and controls' nondominant arm was inconclusive (t=−2.1, $p_{tukey}$=0.1, $BF_{10}$=1.2). Similar results

were obtained when testing for the unique effect of noise beyond bias, by adding artificial arm bias as a covariate when comparing the motor noise of the artificial arm errors between the three groups (see *Supplementary file 1c* for a full statistical report). These results show that the congenital group's artificial arm reaches are best characterized by increased noise (endpoint variability). In the next analyses, we will test two potential sources of noise: artificial arm sense of localization (proprioception) and adequacy of motor planning and its execution.

## The congenital and acquired groups are equally accurate at localizing their artificial arm without visual feedback

Commercially available artificial arms, as the ones used by our participants, currently lack direct sensory feedback, and of most relevance to reaching, proprioceptive feedback. Proprioception, the sense of position and movement of our body, provides an essential input to the sensorimotor system (*Sarlegna and Sainburg, 2009*; *Wolpert et al., 1995*). Together with vision, it is used to accurately localize the current position of the arm and guides corrective movement during the execution of reaching movements. The lack of proprioception is therefore a reasonable candidate for explaining the inferior control of the artificial arm. As a proxy measure for proprioception, we assessed artificial arm localization abilities. First, to assess artificial arm localization, at its most basic and simple form, we tested artificial arm localization along a single axis. In a separate task, participants were asked to place their artificial arm in an opaque tube and use their intact arm to point to the endpoint of the artificial arm (*McDonnell et al., 1989*; see Materials and methods). Their entire affected upper limb (residual and prosthesis) was covered and not visible during this task. Since the artificial arm is sensed and localized via the residual limb, we also assessed the proprioception of the residual limb. Interestingly, we found no localization differences between the congenital and acquired groups in either condition (Residual limb: Mann-Whitney W=112.5, p=0.46, $BF_{10}$=0.26; Artificial arm: Mann-Whitney W=104.5, p=0.7, $BF_{10}$=0.24; see *Figure 2C*), suggesting that individuals with a congenital limb difference are equally able as individuals with an acquired limb difference at localizing their artificial arm.

While artificial arm localization does not seem to differ between our two limbless groups, it is still possible that the online integration of localization input, rather than the input itself, is suboptimal in the congenital group. To test this, we asked participants to perform a 2D localization task. This task was very similar to our main reaching task, with the exception that participants reached to visual targets without receiving continuous visual feedback of their limb position (see Materials and methods). Here, participants were instructed to prioritize accuracy in their performance, that is, were allowed to make corrective movements and were not limited in their movement time. Overall, participants made larger errors and took longer to get to the target (movement time) in this task compared to the main task (see *Supplementary file 1d&e* for full statistical analysis showing a significant task effect for both measures). Using the same ANCOVA approach described above, we compared artificial arm (and nondominant arm) errors of the three groups, while controlling for intact-hand performance as a covariate. We found no group differences in artificial arm errors ($F_{(2,44)}$=0.71, p=0.5, $BF_{Incl}$=0.33; see *Figure 2D*). We further performed a planned pairwise comparison between the artificial arm performance of the congenital and acquired groups and found no significant difference (t 0.71, $p_{unc}$=0.48, $BF_{10}$=0.25). Taken together, these results suggest that the congenital group's artificial arm localization is not substantially different than that of amputees.

## The congenital group shows larger artificial arm initial directional errors

Fast reaching movements, such as the ones performed in our main task, can be roughly divided into two phases: an *initial impulse phase* that involves the execution of a motor plan constructed prior to movement initiation and an *error correction phase* where sensory information is used to correct errors during execution (*Elliott et al., 2001*). The timing of peak velocity in such a movement is often used as a time point that mostly reflects the first phase, that is, the trajectory up to this point is mostly governed by feedforward mechanisms (*Krakauer et al., 1999*; *Patterson et al., 2017*; *Sainburg et al., 2003*). As the error at the end of the reach can originate from both feedforward processes and sensory integration processes, comparing the errors at the initial phase allows us to disentangle feedforward from feedback mechanisms. Specifically, the directional error at this stage provides a measure of how far away the movement is from the target's direction. To measure the initial directional error for each trial we took the direction vector (the line connecting the home position and the arm's location

at peak velocity; see *Figure 2B*) and calculated the absolute angle between that direction vector and the target direction vector (the line connecting the target and the home position). We use the initial directional error to characterize the early part of the arm trajectory as an indicator of the accuracy of the motor plan. As with previous measures, initial directional errors were analyzed using an ANCOVA comparing directional errors of the artificial arm (and nondominant arm) of the three groups, while controlling for intact-hand directional errors as a covariate. We found significant group differences in artificial arm errors ($F_{(2,47)}$=8.01, p<0.001, $\eta_p^2$=0.26). The congenital group exhibited larger directional errors with their artificial arm compared to the acquired group (t=−3.515, p=0.003, Cohen's-d=−1.10) and the control group (t=−3.31, p=0.005, Cohen's-d=−0.98). The acquired group's artificial arm directional errors did not differ from those of the nondominant arm of the control group (t=0.16, p=0.9, $BF_{10}$=0.21). This result suggests that the initial motor plan of the congenital group differs from that of the acquired and control groups. However, based on this measure alone, we are unable to distinguish between errors resulting from the motor plan itself or with noise resulting from its execution.

As a complementary approach, we also compared the angle of the corrective movement, which follows the initial movement. While participants were instructed to perform a single reaching movements, and to avoid secondary corrective movements, this later phase of the movement is still known to be influenced by sensory feedback (*Elliott et al., 2001*), and thus may provide additional insight into the contributing factors to the absolute reaching error. For each trial, we have calculated the absolute angle between the direction vector at peak velocity and the direction vector at the end of the movement. A perfectly straight reach (with no corrective movements) would therefore have a corrective angle of 0. Using the same ANCOVA approach described above, we found inconclusive evidence for group differences in artificial arm corrective angles ($F_{(2,47)}$=2.19, p=0.12, $BF_{Incl}$=0.67). Within the congenital group, corrective angles were highly correlated with initial error angle ($r_{(16)}$=0.76, p<0.001) and not correlated with motor variability ($r_{(16)}$=0.35, p=0.16, $BF_{10}$=0.73). In other words, individuals who made larger initial errors had more to correct for and therefore showed bigger corrective movements. However, individuals who made larger corrective movements were not necessarily more accurate in reaching the target. Given this inherent dependency, this measure does not provide us with an opportunity to decouple the contribution of the feedback correction from the size of the feedforward error.

## Early life but not present experience with artificial arms affects adult motor control in individuals with a congenital limb difference

We tested the hypothesis that present artificial arm usage will have a significant relationship on users' artificial arm motor control. First, we confirmed that although the congenital group has accumulated on average ~29 more years of (intermittent) artificial arm experience compared to the acquired group ($t_{(30)}$=−7.86, p<0.001), there are no differences in artificial arm daily usage between the two artificial arm users' groups, as assessed using questionnaires (acquired vs. congenital groups; $t_{(30)}$=−0.25, p=0.81, $BF_{10}$=0.35). Contrary to our hypothesis, we found no such relationship between artificial arm reaching errors and a daily life artificial arm usage score, encompassing both daily wear-time and functionality of use ($r_{(30)}$=−0.05, p=0.78, $BF_{10}$=0.23). We did, however, find a relationship between daily life artificial arm usage and intact-hand reaching errors across acquired and congenital groups ($r_{(39)}$=−0.41, p=0.008; see *Figure 1—figure supplement 1*). Smaller intact-hand errors (higher accuracy) were associated with higher artificial arm use scores (more versatile and frequent use). While being cautious not to infer causality from a correlation, we believe this result uncovers a relationship between an individual's general motor control (as measured by their intact arm) and their ability to use an artificial arm. This result further highlights the need to control for individual differences in intact-hand motor control when studying artificial arms.

While we found that artificial arm present-day use does not predict its motor control (i.e., absolute error in reaching), we wanted to next test whether the user's early life experience does. Quantifying something as complex as artificial arm past use is a difficult feat. Here, we focused on the age at which an individual with a congenital limb difference was first fitted with an artificial arm (range: 2 months to 11 years). Interestingly, we found a significant positive correlation between age at first artificial arm use and artificial arm reaching errors ($r_{s(16)}$=0.57, p=0.01; see *Figure 1D*). Individuals who started using an artificial arm at an earlier age produced smaller errors with their current artificial arm as adults. This result suggests that our ability to adjust our motor representation might not be as flexible as

we thought and might be constrained by early life experience. Finally, we did not find a significant relationship between past (age at first artificial arm use) and present (daily life use score) artificial arm experience ($r_{s(16)}$=−0.03, p=0.91) or between artificial arm reaching errors and elapsed time since first artificial arm use ($r_{(16)}$=0.01, p=0.96, $BF_{10}$=0.29).

### Across the congenital group, age at first experience with an artificial arm correlates with motor noise

Next, we wanted to test which of the aforementioned measures, each representing a different aspect of motor control, best correlates with age at first artificial arm use. Discovering which of these measures are influenced by age at first artificial arm use can give us an insight into which age-constrained motor control process might be involved in learning to control an artificial limb. Motor noise and initial directional error, being the two measures that produced a significant group effect, are of special interest, but for the sake of completeness, we have tested the potential impact of all six measures. We found that only motor noise significantly correlated with age at first artificial arm use ($r_{s(16)}$=0.73, $p_{bonf}$=0.003 [corrected for six comparisons]; see *Figure 2E*). That is, individuals who started using an artificial arm earlier in life also showed less endpoint motor noise in the reaching task. While we did find group differences in initial directional errors (as reported in The congenital group shows larger artificial arm initial directional errors), it did not significantly correlate with age at first artificial arm use ($r_{s(16)}$=0.38, $p_{bonf}$=0.74). From these results, we infer that early life experience relates to a suboptimal ability to reduce the system's inherent noise, and that this is possibly not related to the noise generated by the execution of the initial motor plan. Early life experience might therefore relate to better use of visual feedback in performing corrective movements. The continuous integration of visual and sensory input is at the core of visually driven corrective movements. Therefore, one possibility is that limited early life experience, results in suboptimal integration of visual and sensory information within the sensorimotor system.

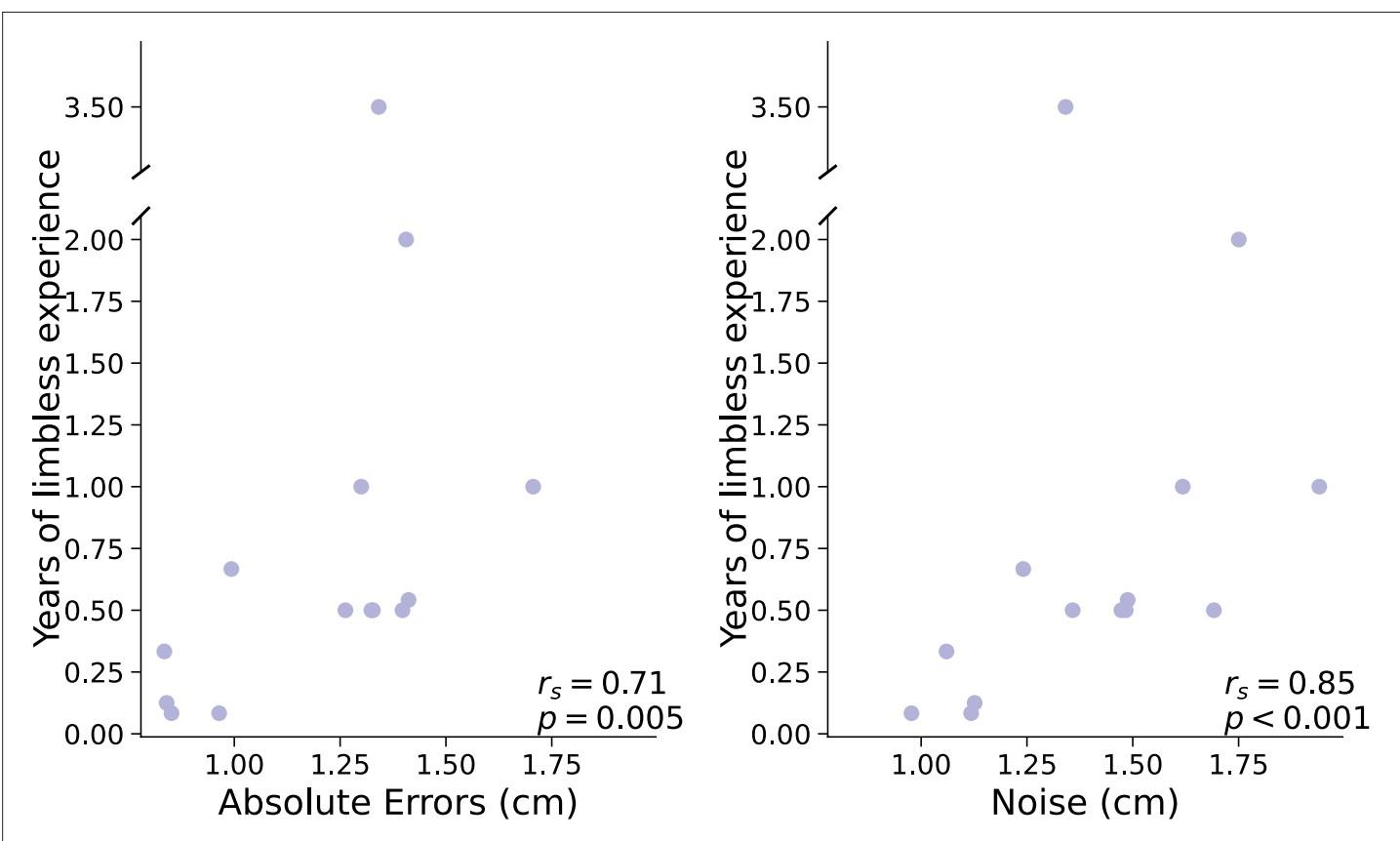

**Figure 3.** Years of limbless experience before first artificial arm use in the acquired group. (**A**) Relationship between years of limbless experience and (**A**) artificial arm reaching errors or (**B**) artificial arm motor noise in the acquired group.

## Artificial arm control of the acquired group might also be constrained by a time-sensitive process following amputation

Finally, we wanted to explore whether the acquired group also shows a parallel phenomenon of an effect of early life experience of artificial arm use on current motor control. From the relationship observed in the congenital group, we can draw two predictions with regards to the acquired group: first, that the age at which you started using an artificial arm (even in adulthood) would potentially have an effect on reaching accuracy. So, the younger you were when you learned to use an artificial arm the better. However, we found no such correlation between artificial arm errors and age at first artificial arm use in the acquired group ($r_{s(12)}$=−0.23, p=0.43).

An alternative parallel to the age at first artificial arm use in the acquired group is the amount of time an individual has spent being limbless before starting to use an artificial arm. So, the longer one waits after amputation to start using an artificial arm, the bigger their reaching errors would be. Here, we find a significant positive correlation between artificial arm absolute errors and years of limbless experience ($r_{s(12)}$=0.71, p=0.005; see *Figure 3*). The sooner an individual with an acquired limb difference was fitted with an artificial arm after amputation, the smaller errors they made with their current artificial arm years later. Similar to the congenital group, this relationship appears to be driven by motor noise and not by bias. Motor noise was significantly correlated with years of limbless experience ($r_{s(12)}$=0.85, $p_{bonf}$=0.03; see *Figure 3*), while bias did not ($r_{s(12)}$=0.32, $p_{bonf}$=1; comparing between correlations: Z=2.9, p=0.003). This suggests that the link between age of first use and errors in the congenital group may not be limited to a developmental period, but to an individual's first experiences as a limbless individual. Alternatively, this finding points toward a possible plasticity window in the time after amputation, where early exposure to an artificial arm results in higher levels of control. Although the type of plasticity bottlenecks in each group might be different, it appears that the amount of time an individual spends using their residual limb before starting to use an artificial arm has a long-lasting effect, in terms of motor noise, on their ability to control an artificial arm.

## Discussion

While infants' reaches are surprisingly functional (*Babinsky et al., 2012*; *von Hofsten, 1980*), they take a considerable amount of time to be fine-tuned. There are multiple, non-linear (*Olivier et al., 2007*) developmental processes occurring until at least the age of ~12 years (*Schneiberg et al., 2002*; *Simon-Martinez et al., 2018*; *Sveistrup et al., 2008*). In this context, it may not be surprising that we found that individuals with a congenital limb difference, who started using an artificial arm in early childhood, performed differently in a visuomotor precision reaching task, relative to individuals with an acquired limb difference, who only began to use an artificial arm in adulthood (age range=19–56, mean=32). Yet, contrary to our expectation, the congenital group performed worse than the acquired group, who in turn did not show any deficits in controlling their artificial arm relative to two-handed controls using their nondominant arm. Considering the early artificial arm experience of the congenital group, and that it coincides with a time in development when the motor system has to constantly adapt as the body (and arms) grow, it is surprising that the congenital group under-performs at this straightforward motor task relative to the acquired group. The observed difference between the two artificial arm user's groups, and the fact that increased experience with an artificial arm did not lead to better performance, is also in stark contrast with the notion that our internal model flexibly scales with the endpoint of the tool, as we use it (*Miller et al., 2018*).

What mechanisms might be driving the observed deficits in the congenital group? To successfully position the artificial arm at the visual target, multiple internal calculations and transformations need to occur, each of which could potentially be impacted by early life experience. First, an internal model of the artificial arm needs to be developed or adapted, so as to translate a desired goal into an appropriate motor coordination plan. Our analysis points at potential deficits in this internal model, as the congenital group's initial directional error—reflecting the execution of an initial motor plan and thought to precede sensory feedback—is greater than that of the acquired group. If true, this suggests that individuals with a congenital limb difference may not be able to refine their model enough to create an accurate template of their artificial arm. Another interpretation of this result would be that these individuals have a noisy action-selection process (*Adams et al., 2020*). However, since this deficit was dissociated from our key measure of absolute reaching error, we believe other mechanistic deficits

might be at play. With that in mind, a further step for executing a successful reach is being able to integrate concurrent input from the executed plan with online visual feedback of the artificial arm, as well as any other relevant somatosensory feedback from the residual arm. When performing reaching movements without visual feedback (2D localization task), the congenital group did not differ from the acquired or control group. This begs the question: if the congenital group has a deficit in motor planning, why was it not evident in this task as well? In the 2D localization task, unlike the main task, participants were allowed to make corrective movements. While they did not receive visual feedback, the proprioceptive and somatosensory feedback from the residual limb appears to be enough to allow them to correct for initial reaching errors and perform at the same level as the acquired and control groups. Moreover, we did not find strong evidence for an impaired sense of localization of either the residual or the artificial arm in the congenital group. As such, by elimination, our evidence suggests that the process of using visual information to perform corrective movements is not as efficient in the congenital group. We note that we were unable to capture this potential deficit when focusing solely on the corrective phase of the movement, likely due to the strong coupling with the initial movement phase. Nevertheless, this idea is consistent with previous evidence showing individuals with a congenital limb difference have impaired processing of visual hand information (*Maimon-Mor et al., 2020b*). This interpretation is also compatible with previous studies in individuals experiencing a brief period of postnatal visual deprivation which caused long-lasting (though mild) alterations to visuo-auditory processing (*de Heering et al., 2016*). While the maturation of the vision system occurs much earlier than that of motor control, the ability to optimally integrate visual information continues to develop way into childhood (*Contreras-Vidal, 2006*; *Contreras-Vidal et al., 2005*). Lack of concurrent visual and motor experience during development might therefore cause a deficit in the ability to form the computational substrates and thus to efficiently use visual information in performing corrective movements. Indeed, we found that endpoint noise, and not initial directional error, associates with age at first artificial arm use. This, too, supports the idea that one's ability to efficiently integrate sensory information with motor control might relate to early life experience with an artificial arm.

Perhaps our most intriguing result relates to a relationship between the deficits in motor control (reaching error) and the age in infancy at which individuals with a congenital limb difference started using an artificial arm. Individuals who started using an artificial arm for the first time earlier in infancy also showed less motor deficit. The detected relationship between early life development and motor skill in adulthood allows us to address questions about the plasticity of visuomotor control across life. Why would a 4-year-old child have a disadvantage in visuomotor learning relative to a 2-year-old? This could be explained both by how early she picked up artificial arm use, or rather, how late she waited before starting to use it. While the two alternatives sound similar, these two complementary explanations can be mechanistically dissociated. The first explanation is that the more experience you have with an artificial arm in early childhood, the better you will be at controlling the artificial arm as an adult. Based on the well-accepted assumption that the brain is more plastic early in life, this will allow children to acquire the new skill more easily (*Knudsen, 2004*). Alternatively, templates for motor control of the hand (e.g., driven by genetics) will decay over time if not consolidated by relevant experience-related input (*Dempsey-Jones et al., 2019b*; *Krubitzer and Prescott, 2018*). The longer one waits before including the artificial arm as part of their motor repertoire, the less she will be able to take advantage of this genetic blueprint, that is, in terms of brain structure and function (*Sur and Rubenstein, 2005*). Another, third, explanation relates to the fact that by not wearing an artificial arm, individuals with a congenital limb difference develop alternative strategies to compensate for their missing hand, for example by using their residual limb. Individuals with a congenital limb difference are known to be proficient residual limb users, and our previous research shows that the residual arm benefits from the sensorimotor territory normally devoted to the hand (*Hahamy et al., 2017*; *Makin et al., 2013*). We also previously showed that residual limb use in these individuals impacts larger-scale network organization in sensorimotor cortex (*Hahamy et al., 2015*) demonstrating how compensatory strategies can affect neural connectivity and dynamics. In an extreme scenario, using the residual limb as an effector in early life might anchor it as the reference frame for all upper-limb motor control. We previously found that this has implications on peri-personal space representation in individuals with a congenital limb difference (*Maimon-Mor et al., 2017*). Thus, the later an individual with a congenital limb difference starts wearing their artificial arm, the later they start developing an alternative reference frame, that is, learning the computations and transformations required to

perform movements with an end effector at the artificial arm tip instead of the residual limb. Another way to think about this competition between alternative strategies is through the prism of 'habits' and the idea that once you have perfected a particular motor solution, it is more difficult to update it to a different strategy. As these multiple mechanisms are not mutually exclusive, it is possible that they all contribute to the observed relationship between age at first artificial arm use and reaching errors. Regardless of the specific mechanism, if the congenital group's sensorimotor processes are optimized for treating the tip of the residual limb as the end effector, then when wearing an artificial arm, they constantly required to transform information from the residual limb tip to the artificial arm tip and vice-versa, for extrapolating where the residual limb needs to be to get the artificial arm tip to a certain target. Similar to skill acquisition of tools, this extra step of transforming information between two spatially removed endpoints may introduce additional noise (integration over space and time).

The acquired group, who were born with a complete upper-limb and lost it later in life, did not show similar deficits in artificial arm reaching control. Following the rationale outlined above, it stands to reason that having developed an internal model of their arm in childhood, individuals with an acquired limb difference are able to recycle the internal model of their missing arm to accurately control the artificial arm. Indeed, there is mounting evidence to suggest that individuals with an acquired limb difference maintain the representation of their missing hand long after amputation (*Kikkert et al., 2016*; *Wesselink et al., 2019*). The motor requirements for control over an artificial arm are not identical to that of controlling a biological arm, however, from the perspective of the spatial location of the endpoint, the artificial arm roughly mimics that of the missing arm. Inter-estingly, despite not showing a group deficit in control of their artificial arm (relative to controls' nondominant arm), we still found that the time they have taken to use the artificial arm following their amputation co-varies with artificial arm reaching errors. As with the congenital group, this relationship could be explained both by how early one picks up artificial arm use, or how late she waits before starting to use it. For example, research in stroke patients suggested that the imbal-ance triggered by the assault to the brain tissue creates conditions that are favorable for plasticity, and even been referred to as a second sensitive period (*Zeiler and Krakauer, 2013*). According to this notion, rehabilitation will be most effective within this limited period of plasticity. Similarly, it has been suggested that sensory deprivation can also promote plasticity and learning (*Dempsey-Jones et al., 2019a*). Therefore, amputation might cause a cascade that will result in a brief period of increased plasticity that will be more favorable for learning to use an artificial arm. Alternatively, we can also consider the competition model described above; if one has already formed a motor strategy (or a habit) for how to perform tasks without the artificial arm, this learned strategy will impact her ability to use the artificial arm later on in life. In both the congenital and acquired groups, artificial arm reaching motor noise correlated with the amount of time they spent using only their residual limb. It is therefore tempting to link these two results under a unifying interpretation; however, this requires further research, considering the neural differences between the two groups. While the reported correlation in the acquired group relies on a smaller sample size and thus should be taken with a little more caution, the fact that overall, the acquired group does not show a system-atic deficit relative to the control group, indicates that early life experience drives the observed visuomotor deficits in reaching reported here.

To conclude, the fact that the congenital group shows inferior artificial arm performance compared to the acquired group is surprising considering both the vast capacity for motor learning that humans exhibit and the fact that the congenital group have had more experience with an artificial arm and from a much younger age. By the process of elimination, we have nominated suboptimal visual feedback-based corrections to be the most likely cause underlying this motor deficit. However, more work testing this suggested mechanism directly needs to be carried out to consolidate our interpreta-tion. Moreover, we found that early life experience with an artificial arm (in the case of the congenital group) or early artificial arm experience following amputation (in the case of amputees) has a measur-able effect on artificial arm motor precision in adulthood. In our data set, early life experience with an artificial arm was not a good indicator for successful current artificial arm adoption, however our limited sample size and inclusion criteria of only including individuals who currently use an artificial arm prevents us for making direct clinical recommendations. The relationship between intact-arm performance and current artificial arm use in both the acquired and congenital groups is also of interest to artificial arm rehabilitation and should be taken into account in future studies. In addition,

our research provides insight about the neurocognitive bottlenecks that need to be considered when developing future assistive and augmentative technologies.

## Materials and methods

### Participants

Forty-four artificial arm users were recruited for this study: 21 individuals with a unilateral acquired limb difference following amputation (mean age±std=48.67±12.9, 18 males, 12 with intact right arm), and 23 individuals with a congenital unilateral upper-limb limb difference (transverse deficiency; age±std=46.09±11.22, 11 males, 17 with intact right arm; see *Table 1* for full demographic details). Sample size was based on recruitment capacities considering the unique populations we tested. Seven participants were excluded from all analyses: four participants with a trans-humeral limb loss (three amputees) were not able to perform the tasks with their artificial arm. Two participants (one amputee) had trouble controlling the robotic handle with their artificial arm and therefore their artificial arm reaches data in both tasks has been excluded. Data from all tasks involving the robotic manipulandum of one participant (congenital group) was excluded, due to technical issues with the robotic device.

Additionally, 20 age, gender, and handedness matched two-handed controls were recruited for this study (mean age±std=42.55±15.5, 11 males, 14 with a dominant right arm). For all analyses, the controls' dominant arm was compared to the intact arm of the artificial arm users, and the nondominant arm was compared to the artificial arm. For the sake of brevity, we refer to the dominant arm of controls as the intact-hand and the nondominant arm as the artificial arm.

Participants were recruited to the study between October 2017 and December 2018, based on the guidelines in our ethical approvals (UCL REC: 9937/001; NHS National Research Ethics service: 18/LO/0474), and in accordance with the Declaration of Helsinki. The following inclusion criteria were taken into consideration during recruitment: (1) 18–70 years old, (2) MRI safe (for the purpose of other tasks conducted in the scanner), (3) no previous history of mental disorders, (4) for both of the prosthesis users groups, owned at least one type of prosthesis during recruitment, and (5) for acquired amputees, amputation occurred at least 6 months before recruitment. All participants gave full written informed consent for their participation, data storage, and dissemination.

### Main task

#### Experimental setup

Participants sat in front of the experimental apparatus on a barber-style chair, with their head leaning against a forehead rest. Participants performed horizontal plane reaches while holding a handle of a robotic manipulandum with either their hand or the artificial arm, with the arm strapped to an armrest. A monitor displaying the task was viewed via a mirror, such that participants did not have direct vision of their arm. To further block any vision of the participant's limb a black barber's cape was used to cover their entire upper body, including their elbow and shoulder. Continuous visual feedback of the robotic handle's position (i.e., the intact/artificial arm position) was delivered as a 4 cm diameter white cursor (representing the handle size) with a 0.3 cm diameter circle at the center. The handle's position was recorded with a sampling frequency of 200 Hz.

#### Experimental design—main task

Participants were asked to reach to visual targets while receiving visual feedback of their hand position using each of their arm. To ensure the setup was optimized for artificial arm reaches, participants performed the task with their nondominant/artificial arm first. For each arm, the task began with a set of six practice trials (using targets not included in the task target set), presented to the participant. Performance (accuracy) during these trials was not monitored, however, in cases where the participant did not fully follow the instructions (e.g., participant started moving before the beginning of the trial) or the participants requested more practice, the six practice trials were repeated again, following necessary adjustments.

A trial was initiated once participants placed the cursor within a white square (1.5 cm×1.5 cm) indicating the home (start) position (denoted as position [0,0]). Participants were situated so the home position was aligned with their midline. In each trial, participants reached to a visual target (1.5

**Table 1.** Demographic details of all participants.

Participant: ALD=individual with an acquired limb difference following amputation, CLD=individual with a congenital limb difference, CO=two-handed control; participants marked with an asterisk have valid data only for their intact-hand and were therefore excluded from most analyses. Y since amp=years since amputation. Gender: M=male, F=female. Amp side=side of limb loss or nondominant side: L=left, R=right. Amp level=level of limb loss: TR=trans-radial, TH=trans-humeral. Artificial arm type=preferred type of artificial arm: Cos=cosmetic, Mech=mechanical, Myo=myo-electric. Artificial arm wear time=typical number of hours artificial arm worn per week. PAL=functional ability with an artificial arm as determined by PAL questionnaire (0=minimum function, 1=maximum function). Usage score=Artificial arm usage score combining wear time and PAL. Age at first artificial arm use=Age at which individuals with a congenital limb difference were first introduced to an artificial arm. Years of limbless experience=Time after amputation at which amputees were first introduced to an artificial arm. Residual limb length=measured in cm.

| Participant | Age | Y since amp | Gender | Amp side | Amp level | Amp cause | Artificial arm type | Artificial arm wear time | PAL | Usage score | Age at first artificial arm use | Years of limbless experience | Residual limb length |
|---|---|---|---|---|---|---|---|---|---|---|---|---|---|
| ALD01 | 58 | 14 | M | L | TR | Trauma | Myo | 119 | 0.5 | 1.87 | | 0.5 | 15 |
| ALD02 | 46 | 16 | F | L | TR | Trauma | Myo | 56 | 0.59 | 0.49 | | 0.5 | 14 |
| ALD03* | 50 | 3 | F | L | TR | Trauma | Mech | 77 | 0.44 | 0.44 | | | 18 |
| ALD04* | 53 | 34 | M | L | TH | Trauma | Mech | 48 | 0.2 | −1.4 | | 0.33 | |
| ALD05 | 21 | 1 | M | R | TR | Trauma | None | 0 | 0.04 | −3.44 | | 0.083 | 34 |
| ALD06* | 42 | 18 | M | R | TR | Trauma | Cos | 35 | 0.07 | −2.33 | | | 16 |
| ALD07 | 61 | 21 | M | L | TR | Trauma | Cos | 105 | 0.67 | 2.21 | | 0.125 | 29.5 |
| ALD08 | 60 | 42 | M | R | TR | Trauma | Mech | 87.5 | 0.28 | 0.05 | | 3.5 | 9 |
| ALD09* | 65 | 37 | M | R | TH | Trauma | Mech | 98 | 0.46 | 1.11 | | 0.25 | |
| ALD10* | 47 | 21 | M | R | TH | Trauma | Cos | 84 | 0.3 | 0.03 | | 1 | |
| ALD11 | 68 | 12 | M | L | TR | Trauma | Mech | 35 | 0.54 | −0.31 | | 0.33 | 21.5 |
| ALD12 | 49 | 5 | M | R | TR | Vascular disease | Cos | 42 | 0.59 | 0.1 | | 0.5 | 20 |
| ALD13 | 57 | 29 | M | L | TR | Trauma | Mech | 65 | 0.11 | −1.31 | | 1 | 18 |
| ALD14 | 53 | 33 | M | L | TR | Trauma | Myo | 98 | 0.43 | 0.98 | | 0.67 | 12 |
| ALD15* | 28 | 10 | F | R | TR | Trauma | Mech | 2 | 0 | −3.55 | | 5 | 7.5 |

*Table 1 continued on next page*

*Table 1 continued*

| Participant | Age | Y since amp | Gender | Amp side | Amp level | Amp cause | Artificial arm type | Artificial arm wear time | PAL | Usage score | Age at first artificial arm use | Years of limbless experience | Residual limb length |
|---|---|---|---|---|---|---|---|---|---|---|---|---|---|
| ALD16 | 29 | 11 | M | L | TR | Trauma | None | 0 | 0 | −3.61 | | 2 | 28.5 |
| ALD17 | 43 | 20 | M | R | TR | Trauma | Myo | 98 | 0.61 | 1.75 | | 0.083 | 8 |
| ALD18 | 55 | 12 | M | L | TR | Trauma | Myo | 98 | 0.65 | 1.92 | | 0.5 | 33 |
| ALD19 | 61 | 17 | M | L | TR | Trauma | Mech | 91 | 0.74 | 2.11 | | 1 | 18 |
| ALD21 | 30 | 3 | M | L | TR | Trauma | Myo | 49 | 0.59 | 0.29 | | 0.54 | 20.5 |
| CLD01 | 51 | | F | L | TR | Congenital | Cos | 7 | 0.26 | −2.3 | 0.5 | | 10 |
| CLD02 | 47 | | M | L | TR | Congenital | Mech | 84 | 0.7 | 1.75 | 1 | | 13 |
| CLD03 | 45 | | F | L | TR | Congenital | Myo | 63 | 0.46 | 0.13 | 4.5 | | 8 |
| CLD04* | 26 | | M | L | TR | Congenital | Mech | 6 | 0.13 | −2.88 | 0.25 | | 15 |
| CLD05* | 55 | | F | L | TR | Congenital | Cos | 112 | 0.3 | 0.82 | 0.25 | | 6 |
| CLD06 | 63 | | M | L | TR | Congenital | Cos | 87.5 | 0.35 | 0.35 | 2 | | 10 |
| CLD07 | 35 | | M | L | TR | Congenital | Cos | 56 | 0.28 | −0.84 | 3 | | 11 |
| CLD09 | 49 | | M | L | TR | Congenital | Myo | 91 | 0.57 | 1.39 | 2 | | 21 |
| CLD10 | 42 | | M | L | TR | Congenital | Cos | 56 | 0.54 | 0.28 | 2 | | 10.5 |
| CLD11 | 66 | | F | R | TR | Congenital | Cos | 42 | 0.35 | −0.93 | 5 | | 9 |
| CLD12 | 56 | | F | R | TR | Congenital | Cos | 98 | 0.43 | 0.98 | 3 | | 11.5 |
| CLD13* | 53 | | M | L | TH | Congenital | Mech | 63 | 0.33 | −0.43 | 2 | | |
| CLD14 | 42 | | M | L | TR | Congenital | Mech | 2 | 0.09 | −3.17 | 4 | | 12 |
| CLD15 | 55 | | F | L | TR | Congenital | Myo | 105 | 0.65 | 2.12 | 3 | | 11.5 |
| CLD17 | 29 | | M | L | TR | Congenital | Myo | 70 | 0.46 | 0.33 | 1 | | 12 |
| CLD18 | 53 | | F | L | TR | Congenital | Cos | 48 | 0.65 | 0.52 | 1.5 | | 7 |

*Table 1 continued on next page*

*Table 1 continued*

| Participant | Age | Y since amp | Gender | Amp side | Amp level | Amp cause | Artificial arm type | Artificial arm wear time | PAL | Usage score | Age at first artificial arm use | Years of limbless experience | Residual limb length |
|---|---|---|---|---|---|---|---|---|---|---|---|---|---|
| CLD20 | 52 | | F | R | TR | Congenital | Myo | 32.5 | 0.26 | −1.58 | 0.3 | | 11.5 |
| CLD21 | 32 | | F | R | TR | Congenital | Myo | 40 | 0.41 | −0.73 | 0.5 | | 9 |
| CLD22 | 57 | | M | R | TR | Congenital | Mech | 126 | 0.69 | 2.88 | 2 | | 15.5 |
| CLD23 | 47 | | F | L | TR | Congenital | Myo | 84 | 0.89 | 2.56 | 11 | | 8.5 |
| CLD25 | 41 | | M | L | TR | Congenital | Myo | 112 | 0.85 | 3.17 | 0.25 | | 8 |
| CO01 | 28 | | F | L | | | | | | | | | |
| CO03 | 40 | | F | L | | | | | | | | | |
| CO04 | 59 | | M | L | | | | | | | | | |
| CO05 | 27 | | F | R | | | | | | | | | |
| CO06 | 61 | | F | L | | | | | | | | | |
| CO07 | 35 | | M | L | | | | | | | | | |
| CO08 | 34 | | F | L | | | | | | | | | |
| CO09 | 24 | | M | L | | | | | | | | | |
| CO10* | 70 | | M | L | | | | | | | | | |
| CO11 | 24 | | M | L | | | | | | | | | |
| CO12 | 18 | | F | L | | | | | | | | | |
| CO13 | 67 | | M | L | | | | | | | | | |
| CO14 | 50 | | M | R | | | | | | | | | |
| CO15 | 51 | | F | L | | | | | | | | | |
| CO16 | 36 | | F | L | | | | | | | | | |
| CO17 | 41 | | M | R | | | | | | | | | |

*Table 1 continued*

| Participant | Age | Y since amp | Gender | Amp side | Amp level | Amp cause | Artificial arm type | Artificial arm wear time | PAL | Usage score | Age at first artificial arm use | Years of limbless experience | Residual limb length |
|---|---|---|---|---|---|---|---|---|---|---|---|---|---|
| CO18 | 33 | | M | R | | | | | | | | | |
| CO19 | 45 | | M | R | | | | | | | | | |
| CO20 | 54 | | M | R | | | | | | | | | |
| CO21 | 53 | | M | L | | | | | | | | | |

cm×1.5 cm) presented in 1 of the 60 predefined locations (see *Figure 1B*). To encourage participants to perform fast-reaching movements, a maximum movement time of 1 s per reach was imposed. At the end of a trial (1 s following movement initiation), the target changed color to blue to indicate the reach has ended and the endpoint position was recorded. To reduce fatigue and experiment duration, participants were then mechanically assisted by the manipulandum moving the handle back to the home position, before the start of the next trial. To quantify participants' biological and artificial arm motor control, participants were asked to perform a single movement to the target and avoid corrective movements. Constant visual feedback of the arm's position was given. Movement initiation was defined by arm velocity exceeding 3.5 cm/s starting from the time of participants' first movement, following the presentation of the target.

## Data Processing and analysis—main task

Within the 1 s movement time constraint, in some trials, participants still performed secondary corrective movements. We therefore used the tangential arm velocities to identify the end of the first reach in each trial (i.e., movement termination). Velocity data were smoothed using an 8 Hz low-pass Butterworth filter (*Przybyla et al., 2013*). Movement termination was defined by the first minimum with a velocity smaller than 50 % of the peak velocity. We note that very similar results were observed when using the end of the trial (1 s after initiation) as the movement termination time. Individual trials were excluded if they were accidentally initiated, that is, if movement terminated close to the home position—closer than 2 cm or with a y-value (depth) smaller than 1 cm; or if the participants did not finish their reach at the end of the allocated time (1 s)—that is, trials where movement >10 cm/s was recorded at the end of the trial. An average of 1.1 and 1.4 trials per participant were excluded for the intact and artificial arm, respectively, with a range of 0–7. There were no group differences in the amount of excluded trials. Artificial arm reaches data from one participant from the acquired group and one participant from the congenital group were excluded, due to technical issues with the device.

Absolute Euclidean error from the target was used as the main measure (see *Figure 1A and C*). Motor noise (variability) and bias were calculated for each participant, for each arm, by aggregating across all targets. Error vectors of each trial (the line connecting the target with the endpoint location) were overlaid as if they were made to a single target (see *Figure 2A*). Bias was defined as the distance from the center of the overlaid endpoints (calculated as the mean x and mean y of the relative error vectors) to the target. Noise was defined as the spatial SD of endpoints relative to the same center of overlaid endpoints. Initial directional error was defined for each trial as the absolute angle between the direction vector—the line connecting the home position and the arm's location at peak velocity (see *Figure 2B*); and the direction vector—the line connecting the target and the home position. The arm's location at peak velocity was used as a proxy for a time point that mostly reflects the motor planning phase, that is, feedforward mechanisms, before sensory information is used to correct for errors during execution. Corrective movements were defined as the absolute angle between the direction vector at peak velocity and the direction vector at the end of the movement.

## Additional tasks

### 1D arm Localization

To assess residual limb and artificial arm sense of limb-position, we used a task similar to that described in *McDonnell et al., 1989*. Participants were asked to place their residual limb or artificial arm in an opaque tube (see *Figure 2C*). In each trial, an adjustable contact plate was placed at a different position within the tube. Participants were asked to move their arm into the tube until it made contact with the plate. They were then instructed to use their intact arm to mark the estimated location of their tested arm on a paper strip placed on the side of the tube. At the end of the trial, participants were asked to remove their arm from the tube before the start of the next trial. A barber's cape was used to cover the upper body and arms. For each condition (residual/artificial arm), pseudo-randomized eight distances were used; these were calculated as a percentage of participant's maximum reach distance (25–75%). The mean absolute distance between the participant's estimate and true position was used as a measure of 1D localization abilities. Two participants from the acquired group and two participants from the congenital group did not take part in this task as it was introduced later in the data collection process.

## 2D arm localization (reaching without visual feedback)

The 2D arm localization task was almost identical to our main task, with the exception that participants reached to visual targets without receiving continuous visual feedback of their limb position and were allowed to perform corrective movements. To prevent a perceptive drift from the lack of visual information of limb position, visual feedback of the arm's position was given at the end of the trial when returning to the start position. The cursor only reappeared when the arm was less than 3 cm away from the start position. At the beginning of the trial, the cursor disappeared upon movement initiation. Movement termination and the recording of the arm's final position occurred when velocity was less than 3.5 cm/s for more than 1 s. Due to the noisier nature of these reaches, each of the 60 targets used in the main task was repeated twice (i.e., a total of 120 trials for each arm).

Absolute Euclidean error from the target was used as the main measure. Individual trials were excluded if they were accidentally initiated (see main task data analysis protocol); or if the trial was suspected as invalid, that is, movement time was longer than 10 s or error was larger than 20 cm. An average of 1.35 and 1.5 trials per participant were excluded for the intact and artificial arm, respectively, with a range of 0–13. There were no group differences in the amount of excluded trials. Two participants only produced partial data, missing artificial arm reaches of one participant from the congenital group and dominant arm reaches of one participant from the control group.

## Artificial arm usage assessment

Participants completed a questionnaire to assess various aspects of their current and past artificial arm use. Frequency and functionality of artificial arm use were combined to create an overall artificial arm usage score (as previously used in *Maimon-Mor et al., 2020c*; *Maimon-Mor and Makin, 2020a*; *van den Heiligenberg et al., 2017*; *Van Den Heiligenberg et al., 2018*). To determine frequency of use, participants were asked to indicate the typical number of hours per day, and days per week, they use their artificial arm. These were then used to determine the typical number of hours per week that the artificial arm was worn. To determine the functionality of artificial arm use, participants were asked to complete the artificial arm activity log (Prosthesis Activity Log—PAL), a modified version of the Motor Activity Log (MAL) questionnaire, which is commonly used to assess arm functionality in those with upper-limb impairments (*Uswatte et al., 2006*). The PAL consists of a list of 27 daily activities (see https://osf.io/jfme8/); participants rated how often they incorporate their artificial arm to complete each activity on a scale of 'never' (0 point), 'sometimes' (one point), or 'very often' (two points). The PAL score is then calculated as the sum of all points divided by the maximum possible score, generating a value between 0 (no functionality) and 1 (maximum functionality). Artificial arm scores were calculated for the most used artificial arm, wear time and PAL were standardized using a Z-transform and summed to create an artificial arm usage score that reflects the frequency of use and incorporation of the artificial arm in activities of daily living. These measures have been previously shown to have good reliability using a test-retest assessment (*Maimon-Mor et al., 2020c*).

The congenital group was asked to report the age at which they first used an artificial arm. Two participants (CLD16 and CLD17) were assigned a value of 1 year after responding: 'months old' and 'less than a year,' respectively. Acquired amputees were asked to report the time after amputation in which they were fitted with their first artificial arm. Two participants (ALD13 and ALD19) were assigned a value of 1 year after responding: 'same year, few months after amputation' and 'a few months after amputation,' respectively.

## Statistical analyses

Statistical analyses were carried out using JASP (*Jasp Team, 2020*). An ANCOVA was used to test group differences for all measures in which we had recorded performance of both arms (i.e., all measures but 1D localization). The artificial arm performance was the dependent variable, intact-hand performance was defined as a covariate and group (control, acquired, and congenital groups) as a between-subject variable. Post hoc group tests were corrected for multiple comparisons (Tukey correction). Absolute error measures were logarithmically transformed and then averaged in order to correct for the skewed error distribution and satisfy the conditions for parametric statistical testing. Outliers were defined as 1.5 times the IQRs (interquartile ranges) below the first quartile or above the third quartile of the transformed error. Following this outlier criteria, in the main task, two participants (one from the acquired group, one from the congenital group) were excluded due to their high artificial arm errors. For the 2D

localization task, three participants (two from the acquired group, one from the control group) were excluded due to their high intact-arm errors. In parametric analyses (ANCOVA, ANOVA, and Pearson correlations), where the frequentist approach yielded a non-significant p-value, a parallel Bayesian approach was used and Bayes Factors (BFs) were reported (*Morey and Rouder, 2015*; *Rouder et al., 2009*; *Rouder et al., 2012*; *Rouder et al., 2016*). A BF <0.33 is interpreted as support for the null hypothesis, BF >3 is interpreted as support for the alternative hypothesis (*Dienes, 2014*). In Bayesian ANOVAs and ANCOVAs, the inclusion BF of an effect ($BF_{Incl}$) is reported, reflecting that the data is X (BF) times more likely under the models that include the effect than under the models without this predictor. When using a Bayesian t-test, a Cauchy prior width of 1.39 was used, this was based on the effect size of the main task, when comparing artificial arm reaches of the congenital and acquired groups. Therefore, the null hypothesis in these cases would be there is no effect as large as the effect observed in the main task. Parametric analyses were used if assumptions (e.g., for normality) were met, otherwise a Spearman correlation/Mann-Whitney were used. Since the Spearman correlation has, to our knowledge, no current Bayesian implementation no BF values are reported for these tests. The Parametric Watson-Williams multi-sample test for equal means was used as a one-way ANOVA test for bias angular data.

## Acknowledgements

This work was supported by an ERC Starting Grant (715022 EmbodiedTech), awarded to TRM, who was further funded by a Wellcome Trust Senior Research Fellowship (215575/Z/19/Z). ROMM is supported by the Clarendon scholarship and University College, Oxford. The author thank Nour Odeh, Mischa Dhar, and Victoria Root for data collection and Dorothy Cowie and Jordan Taylor for their comments on the manuscript. The authors thank Opcare for their help in participants recruitment, and our participants and their families for their ongoing support of our research.

## Additional information

### Competing interests

Tamar R Makin: Senior editor, *eLife*. The other authors declare that no competing interests exist.

### Funding

| Funder | Grant reference number | Author |
| --- | --- | --- |
| H2020 European Research Council | 715022 EmbodiedTech | Tamar R Makin |
| Wellcome Trust | Senior Research Fellowship (215575/Z/19/Z) | Tamar R Makin |
| Clarendon Fund | Graduate Student fellowship | Roni O Maimon-Mor |
| University College, Oxford | Graduate Student fellowship | Roni O Maimon-Mor |
| UK Research and Innovation | UKRI Turing AI Fellowship (EP/V025449/1) | A Aldo Faisal |

The funders had no role in study design, data collection and interpretation, or the decision to submit the work for publication.

### Author contributions

Roni O Maimon-Mor, Conceptualization, Data curation, Formal analysis, Investigation, Methodology, Project administration, Software, Validation, Visualization, Writing – original draft, Writing – review and editing; Hunter R Schone, Data curation, Investigation, Project administration, Validation, Writing – review and editing; David Henderson Slater, Resources; A Aldo Faisal, Conceptualization, Supervision, Writing – review and editing; Tamar R Makin, Conceptualization, Funding acquisition, Resources, Supervision, Writing – original draft, Writing – review and editing

## Author ORCIDs
Roni O Maimon-Mor http://orcid.org/0000-0001-5262-9976
A Aldo Faisal http://orcid.org/0000-0003-0813-7207
Tamar R Makin http://orcid.org/0000-0002-5816-8979

## Ethics
Human subjects: Participants were recruited to the study between October 2017 and December 2018, based on the guidelines in our ethical approvals (UCL REC: 9937/001; NHS National Research Ethics service: 18/LO/0474), and in accordance with the declaration of Helsinki. All participants gave full written informed consent for their participation, data storage and dissemination.

## Decision letter and Author response
Decision letter https://doi.org/10.7554/eLife.66320.sa1
Author response https://doi.org/10.7554/eLife.66320.sa2

---

# Additional files

### Supplementary files
• Supplementary file 1. Supplementary full statistical reports. (a) *Main analysis while controlling for artificial arm/nondominant arm side.* Results of a follow-up ANCOVA analysis showing no effects of artificial arm side (L vs. R) on artificial arm reaching errors. Our main finding of a significant group effect was also unaffected by accounting for the side of the arm making the reaches. (b) *Main analysis while controlling for residual limb length.* Results of a follow-up ANCOVA analysis showing no effects of residual limb length on artificial arm reaching errors. Our main finding of a significant group effect was also unaffected by accounting for residual limb length. Note that this analysis only includes artificial arm users (congenital and acquired) as controls have a complete arm and therefore no residual limb length (c) *Comparing artificial arm error noise while controlling for artificial arm bias.* Results of a follow-up ANCOVA analysis showing that while there is a significant relationship between bias and noise, the group differences in error noise are independent of bias. (d) Analysis of reaching errors comparing the main task and the 2D localization task. Participants made overall larger errors in the 2D localization task compared to the main task which included visual feedback. (e) Analysis of movement time comparing the main task and the 2D localization task. Participants took longer to move to the target in the 2D localization task compared to the main task which included visual feedback.

• Transparent reporting form

### Data availability
All data generated and analysed during this study can be found at https://osf.io/quyke/.

The following dataset was generated:

| Author(s) | Year | Dataset title | Dataset URL | Database and Identifier |
| --- | --- | --- | --- | --- |
| Maimon Mor RO, Makin TR | 2021 | Artificial-arm (prosthesis) motor control | https://doi.org/10.17605/OSF.IO/QUYKE | Open Science Framework, 10.17605/OSF.IO/QUYKE |

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

# Appendix 1

## Speed-accuracy trade-off in artificial arm reaches

To test whether the three groups (control, acquired, and congenital groups) use the same speed-accuracy trade-off strategy. More specifically, whether artificial arm reaches follow the same control principles as biological arm reaches that have been shown to follow Fitts' law (***Fitts, 1954***), a specific relationship between the movement time and the movement distance:

$$\mathrm{MT} = \mathrm{a} + \mathrm{b} \cdot \log_2 (2D/S)$$

[MT: movement time, D: distance to target, S: target size – constant].

For each subject, we used a linear regression to obtain the parameters a and b. To test whether reaches of each group follow Fitts' Law equally, the r-squared value of the regression was compared across all groups as well as the regression parameters (using an ANCOVA, controlling for the parallel measure of the intact hand). To reduce the influence of noisy individual reaches, the reaches have been divided and averaged into six bins, based on their distance from the starting position. We found no group differences in either goodness of fit (**r²**: p=0.84, BF$_{Incl}$=0.167) or fitted parameters (**a**: p=0.31, BF$_{Incl}$=0.347, **b**: p=0.61, BF$_{Incl}$=0.22) between groups, indicating artificial arms reaches follow Fitts' laws and do not differ in their speed-accuracy trade-off strategy (see ***Appendix 1—figure 1***; see ***Appendix 1—table 1*** for full statistical report and https://osf.io/quyke/ for plots of individual participants).

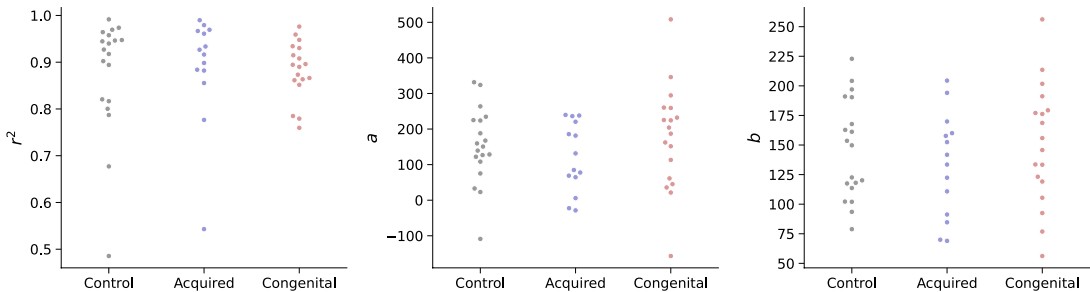

**Appendix 1—figure 1.** Group values for Fitts law model fitting (r², a, b). A linear regression was fit for each participant's reaches to obtain the Fitts law model's parameters a and b. Parameters, as well as goodness-of-fit (r²), were compared across groups. We found no group differences in either goodness of fit (r²: p=0.84, BF$_{Incl}$=0.167) or fitted parameters (a): p=0.31, BF$_{Incl}$=0.347, (b: p=0.61, BF$_{Incl}$=0.22) between groups, indicating artificial arms reaches follow Fitts' laws and do not differ in their speed-accuracy trade-off strategy.

**Appendix 1—table 1.** Frequentist and Bayesian analysis of model fitting reaches data to Fits' Law. Full statistical report of group comparisons of model's parameters a and b as well as goodness-of-fit (r²) of the linear regression model. No differences were found across groups.

| **ANCOVA – r² Artificial arm** | | | | | | **Bayesian ANCOVA** | | | |
|---|---|---|---|---|---|---|---|---|---|
| | | | | | | Analysis of effects – **r² Artificial arm** | | | |
| Factors | SS | df | MS | F | p | Effects | P(incl) | P(incl\|data) | BF $_{incl}$ |
| Group | 0.003 | 2 | 0.002 | 0.175 | **0.84** | Group | 0.5 | 0.143 | **0.167** |
| r² Intact | 0.072 | 1 | 0.072 | 7.587 | 0.008 | r² Intact | 0.5 | 0.854 | 5.852 |
| Residuals | 0.447 | 47 | 0.01 | | | | | | |

*Appendix 1—table 1 Continued on next page*

*Appendix 1—table 1 Continued*

| ANCOVA – $r^2$ Artificial arm | | | | | | Bayesian ANCOVA | | | |
|---|---|---|---|---|---|---|---|---|---|
| ANCOVA – **a** Artificial arm | | | | | | Bayesian ANCOVA | | | |
| | | | | | | Analysis of effects – **a** Artificial arm | | | |
| Factors | SS | df | MS | F | p | Effects | P(incl) | P(incl\|data) | BF $_{incl}$ |
| Group | 27,481.713 | 2 | 13,740.857 | 1.211 | 0.307 | Group | 0.5 | 0.258 | 0.347 |
| a Intact | 158,782.482 | 1 | 158,782.482 | 13.998 | <0.001 | a Intact | 0.5 | 0.982 | 53.771 |
| Residuals | 533,131.773 | 47 | 11,343.229 | | | | | | |
| ANCOVA – **b** Artificial arm | | | | | | Bayesian ANCOVA | | | |
| | | | | | | Analysis of effects – **b** Artificial arm | | | |
| Factors | SS | df | MS | F | p | Effects | P(incl) | P(incl\|data) | BF $_{incl}$ |
| Group | 1378.606 | 2 | 689.303 | 0.498 | 0.611 | Group | 0.5 | 0.178 | 0.217 |
| b Intact | 35,615.379 | 1 | 35,615.379 | 25.73 | < .001 | b Intact | 0.5 | 1 | 3856.606 |
| Residuals | 65,056.014 | 47 | 1384.171 | | | | | | |

