## [Decision Letter]

**Acceptance summary:**

This paper will be of interest to scientists within the field of motor control and for those interested in the development and plasticity of the motor system. The data supports the conclusion of the paper that one-handers reaching accuracy with an artificial arm is smaller than that of amputees, and that in both groups, the performance is related to the number of years experienced before using the artificial arm. However, the data and analyses do not yet provide a clear hypothesis for a mechanistic explanation for the observed results.

**Decision letter after peer review:**

Thank you for submitting your article "Early life experience sets hard limits on motor learning as evidenced from artificial arm use" for consideration by *eLife*. Your article has been reviewed by 2 peer reviewers, and the evaluation has been overseen by a Reviewing Editor and Richard Ivry as the Senior Editor. The reviewers have opted to remain anonymous.

Essential revisions:

1) Please provide more information on the interesting result that the congenital group showed worse reaching ability than the amputees, yet comparable 2D localization abilities. First, it would be helpful to clarify if the 2D localization is simply the reaching task without vision of the cursor? Were the movements of similar duration? Second, it would be helpful to clearly point out that the errors in the 2D localization task are larger than those in reaching-- neither group got better when vision was removed. The two groups simply became more similar. Third, another interpretation that is not clearly spelled out is that this may be due to impaired visually based feedback corrections. Reviewer 1 suggest that an additional analysis would be useful in considering this. See comments below.

2) Reviewer 2 questions the use of the term "one-hander" for the congenital group. I see that you have published with this terminology before. How do people with congenital limb loss refer to themselves and would this label be in any way offensive to them?

3) I wonder how this work fits with the nice study that this group published in 2015 in e*Life*? I am particularly interested in additional thoughts about how neural connectivity would differ between the congenital and amputee groups.

*Reviewer #1 (Recommendations for the authors):*

1. I think analyses of feedback correction will be helpful to understand whether the observed motor noise in one-handers (and some of the amputees) is due to a deficit in feedback responses. This could possibly be done by looking at the change from initial error direction to end point, as well as movement times.

2. Abstract: The concept of visuomotor integration here is very vague, especially when coming in the abstract without mentioning the results that led to this suggestion.

3. Lines 37-50: Does the second hypothesis suggests no difference between one-handers and amputees?

4. Figure 1: The reference to figure 1D comes after referring to figure 2. Consider moving figure 1D elsewhere (or refer to the age analysis sooner).

5. Lines 174-176: It will probably be informative to look at the interaction between target distance and group.

6. Figure 2: Missing individuals' data points.

7. Lines 180 and 182: Should figure references refer to Figure 2B rather than 2A?

8. The targets are quite close to one another. I wonder if there is learning and generalization across targets throughout the experiment.

9. Lines 225-226: Need to clarify here that the residual limb was also not visible during localization.

10. Lines 312-316: The sentences are not clear (improved motor control is related to suboptimal integration): does this mean to say that early experience improves sensory integration (or feedback correction)?

11. Lines 370-374: Does the impaired process of visual hand information could raise the hypothesis that the problem is at a noisy action-selection rather than motor planning?

12. Line 473: missing 'and' between 'arm' and 'therefore'.

13. Line 506-509: what is a "trial set"? and does this mean that there could be more than 6 practice trials, depending on the performance of the participants? If yes, this should be made more clear.

14. Line 514-515: It should be mentioned here how the trial end was determined during the experiment (it is mentioned later, in the data analysis section, saying 1 sec after movement initiation).

*Reviewer #2 (Recommendations for the authors):*

The only other suggestion that I have is that I do not believe the terms used by the authors to refer to their subject populations are appropriate or in keeping with best practices.

I believe "One handers" is entirely inappropriate (and both groups only have one hand, so it is technically ambiguous).

Best-practice is to refer to people and then the condition – so it would be persons with congenital deficiency, or persons with amputation, rather than "one handers" and "amputees".

[Editors' note: further revisions were suggested prior to acceptance, as described below.]

Thank you for resubmitting your work entitled "Early life experience sets hard limits on motor learning as evidenced from artificial arm use" for further consideration by *eLife*. Your revised article has been evaluated by Richard Ivry (Senior Editor) and a Reviewing Editor.

The manuscript has been improved but there is one remaining issue that we would like to see addressed:

Reviewer 1 asks to include the feedback correction analysis in the paper with clear statements about why initial errors may be a confound. This will allow the reader a complete picture of the analysis. To quote the reviewer:

"The authors have successfully addressed almost all of my comments and concerns. The only issue that I'm a bit struggling with is the suggested mechanism that differences between the congenital and acquired groups is a problem with visually-based corrective movements, which has now become a central potential explanation in the paper. I acknowledge that I was the one proposing it, and I'm ok with suggesting it in the paper given the different results between the reaching and 2D localization task. However, in their response letter, the authors included a feedback correction analysis that suggests no difference between the groups, and it seems that they have decided not to include it in the manuscript. I agree that the results might be confounded by initial errors, but I wonder if this should also be included in the paper despite the lack of difference, specifying the potential reasons, and inviting future studies to examine it in a more controlled manner.

In addition, the authors instructed participants to avoid feedback correction in the reaching task, and I think it will be helpful to explain how the interpretation fits with this instruction."

---

## [Author Response]

Essential revisions:1) Please provide more information on the interesting result that the congenital group showed worse reaching ability than the amputees, yet comparable 2D localization abilities. First, it would be helpful to clarify if the 2D localization is simply the reaching task without vision of the cursor? Were the movements of similar duration? Second, it would be helpful to clearly point out that the errors in the 2D localization task are larger than those in reaching-- neither group got better when vision was removed. The two groups simply became more similar.

We apologise for the lack of clarity. These points are now more clearly stated in the text:

Result Section 4 (second paragraph):

“To test this, we asked participants to perform a 2D localisation task. This task was very similar to our main reaching task, with the exception that participants reached to visual targets without receiving continuous visual feedback of their limb position (see Methods). Here, participants were instructed to prioritise accuracy in their performance, i.e., were allowed to make corrective movements and were not limited in their movement time. Overall, participants made larger errors and took longer to get to the target (movement time) in this task compared to the main task (See Supplementary Table 4 and 5 for full statistical analysis showing a significant task effect for both measures).”

Third, another interpretation that is not clearly spelled out is that this may be due to impaired visually based feedback corrections. Reviewer 1 suggest that an additional analysis would be useful in considering this. See comments below.

We agree with this comment – we do not have sufficient evidence to dissociate visually-based feedback corrections from visuomotor integration. We have performed all the analyses requested by Reviewer 1, however the results did not provide the opportunity to tease these two alternative accounts. We therefore revised the manuscript throughout using the terminology ‘visually-based feedback correction’ which we believe describes the behaviour rather than the specific mechanism. For more details, please see response to reviewer’s comments 1.2 and 1.4.

2) Reviewer 2 questions the use of the term "one-hander" for the congenital group. I see that you have published with this terminology before. How do people with congenital limb loss refer to themselves and would this label be in any way offensive to them?

We agree with the reviewer that this is not the best term to identify our participants, and that we should move away from it in future publications. After some discussions with representatives from this community, the preferred term is ‘individuals with a congenital (or acquired) limb difference’. We use this terminology when referring to individuals, and refer to the group as the ‘congenital group’ (in contrast to the amputees which we now refer to as the acquired group).

3) I wonder how this work fits with the nice study that this group published in 2015 in eLife? I am particularly interested in additional thoughts about how neural connectivity would differ between the congenital and amputee groups.

The 2015 study focused on bimanual coordination between the residual-limb and intact hand. Here, we look at finer grained unimanual attributes of artificial (and residual) arm control while regressing out the performance of the intact hand. Moreover, while the connectivity in the 2015 study was found to be related to daily usage habits, here we did not find such a link (with the artificial-arm usage score). When discussing brain plasticity of the residual-arm, we do discuss the 2015 and its relevance to our results.

Discussion (third paragraph)

“Individuals with a congenital limb-difference are known to be proficient residual-limb users, and our previous research shows that the residual arm benefits from the sensorimotor territory normally devoted to the hand (Hahamy et al., 2017; Makin et al., 2013). We also previously showed that residual-limb use in these individuals impacts larger-scale network organisation in sensorimotor cortex (Hahamy et al., 2015) demonstrating how compensatory strategies can affect neural connectivity and dynamics.”

Reviewer #1 (Recommendations for the authors):1. I think analyses of feedback correction will be helpful to understand whether the observed motor noise in one-handers (and some of the amputees) is due to a deficit in feedback responses. This could possibly be done by looking at the change from initial error direction to end point, as well as movement times.

Following the reviewer’s comment, we have performed both analyses. Unfortunately, the statistical tests of these measures were inconclusive. Since the corrective movement angle is confounded by earlier aspects of the movement, we were unable to use them to support any direct claims on impaired visually-based feedback.

Angle of corrective movement:

For each trial, we’ve calculated the absolute angle between the direction vector at maximum speed and the direction vector at the end of the movement. A perfectly straight reach (with no corrective movements) would therefore have a corrective angle of 0. Using the same ANCOVA approach described in the main text, we compared artificial arm (and non-dominant arm) corrective angles of the three groups, while controlling for intact-hand corrective angles as a covariate. We found inconclusive evidence for group differences in artificial arm corrective angles (*F_(_*_2,47)_=2.19, *p*=0.12, BF_Incl_=0.67; see figure below). Within the congenital group, corrective angles were highly correlated with initial error angle (*r*_(16)_=0.76, *p*<0.001) and not correlated with motor variability (*r*_(16)_=0.35, *p*=0.16, BF_10_=0.73). In other words, individuals who made larger initial errors had more to correct for and therefore showed bigger corrective movements. However, individuals who made larger corrective movements weren’t necessarily more accurate in reaching the target. While in principle, this measure had the potential to support our interpretation of a deficit in visual feedback correction. It is unfortunately “contaminated” with the initial reach error making it impossible to decouple the contribution of the feedback correction from the size of the feedforward error.

Movement times:

For each participant, the time between movement initiation and the end of the first reach was extracted and averaged across all trials (See Methods). Longer movement times would theoretically allow for more time for corrective movements based on sensory feedback. When comparing the mean movement time between groups (while controlling for the movement time of the intact hand), we found a trend towards a significant group effect (*F_(2,47)_*=2.94, *p*=0.06, BF_Incl_=1.23). Although the group difference comparison was inconclusive, we have carried out exploratory post-hoc comparisons to be certain that the possible underlying differences do not affect our interpretation, i.e., that the congenital group could not have benefited from a longer time window to integrate the visual information. The congenital group did not significantly differ from either the acquired or control group (*p_tukey_*>0.1 for both). The observed main effect was driven by artificial arm reaches of the acquired group being slightly, but not significantly, faster than controls (*t*=2.335, *p_tukey_*=0.06). As the acquired group did not differ from the control group in reaching performance, unfortunately, this analysis was also not useful in supporting or refuting our interpretation of a deficit in visual feedback correction in the congenital group.

2. Abstract: The concept of visuomotor integration here is very vague, especially when coming in the abstract without mentioning the results that led to this suggestion.

Following the reviewer’s comment, we ran a Liner Mixed Model analysis with all artificial arm reaching trials. We modelled group, distance and group*distance interaction as fixed effects and participant as a random effect. Notice that this model differs from the ANCOVA approach taken in the manuscript as accounting for intact hand performance is less straightforward in this approach. The model revealed a significant distance effect (F(1,47.72)=5.85, p=0.02) and no distance*group interaction (F(2,47.72)=0.54, p=0.59).

3. Lines 37-50: Does the second hypothesis suggests no difference between one-handers and amputees?

Yes, we apologise for not stating this more clearly.

Introduction (third paragraph):

“A second alternative hypothesis is that the congenital group’s early-life disability might offset motor development, but that such disability-related impairment would not necessarily lead to inferior motor performance with an artificial arm. In other words, that the performance of the congenital group would be equivalent to that of the acquired group.”

4. Figure 1: The reference to figure 1D comes after referring to figure 2. Consider moving figure 1D elsewhere (or refer to the age analysis sooner).

We appreciate it may be unconventional but in fact this was intentional. The correlation figure relates to the absolute error measure presented in Figure 1. Moreover, while in terms of narrative we believe the age analysis fits better towards the end of the manuscript, the idea was to have the two main results (takeaways) presented together.

5. Lines 174-176: It will probably be informative to look at the interaction between target distance and group.

Following the reviewer’s comment, we ran a Liner Mixed Model analysis with all artificial arm reaching trials. We modelled group, distance and group*distance interaction as fixed effects and participant as a random effect. Notice that this model differs from the ANCOVA approach taken in the manuscript as accounting for intact hand performance is less straightforward in this approach. The model revealed a significant distance effect (F(1,47.72)=5.85, p=0.02) and no distance*group interaction (F(2,47.72)=0.54, p=0.59).

6. Figure 2: Missing individuals' data points.

Following the reviewer’s comment, for each bar plot presented in Figure 2, we have added a detailed scatter plot in Figure 2- supplement figure 1.

7. Lines 180 and 182: Should figure references refer to Figure 2B rather than 2A?

No, the reference is correct. This section reports the results of the bias-variability analysis which is illustrated in Figure 2A.

8. The targets are quite close to one another. I wonder if there is learning and generalization across targets throughout the experiment.

As our approach focuses on between group differences, any generalisation taking place during the task will occur in all groups and therefor would not affect the interpretation of our results. Nevertheless, it is interesting to consider if the observed differences unfold over time and practice. Following the reviewers comment, we have visualised the cumulative mean error in all groups across the task (see Author response image 1, shaded colours are 95% intervals). In this plot, the value of trial 10 for example, will be the mean of trials 1 to 10. While it appears that error is slightly reducing with time, the relationship in performance between groups remains consistent throughout the task.

**Author response image 1. sa2fig1:** 

9. Lines 225-226: Need to clarify here that the residual limb was also not visible during localization.

Agreed. This has now been specified.

Results (fourth section, first paragraph):

“In a separate task, participants were asked to place their artificial arm in an opaque tube and use their intact arm to point to the end-point of the artificial arm (McDonnell et al., 1989; see Methods). Their entire affected upper limb (residual and prosthesis) was covered and not visible during this task.”

10. Lines 312-316: The sentences are not clear (improved motor control is related to suboptimal integration): does this mean to say that early experience improves sensory integration (or feedback correction)?

We thank the reviewer for pointing this out. These sentences have been rephrased in the revised manuscript.

Result (section 7, first paragraph):

“From these results, we infer that early-life experience relates to a suboptimal ability to reduce the system’s inherent noise, and that this is possibly not related to the noise generated by the execution of the initial motor plan. Early life experience might therefore relate to better use of visual feedback in performing corrective movements. The continuous integration of visual and sensory input is at the core of visually-driven corrective movements. Therefore, one possibility is that limited early life experience, results in suboptimal integration of information within the sensorimotor system.”

11. Lines 370-374: Does the impaired process of visual hand information could raise the hypothesis that the problem is at a noisy action-selection rather than motor planning?

We thank the reviewer for their suggestion, we have mentioned this interpretation in the revised manuscript.

“First, an internal model of the artificial arm needs to be developed or adapted, so as to translate a desired goal into an appropriate motor coordination plan. Our analysis points at potential deficits in this internal model, as one-handers’ initial directional error–reflecting the execution of an initial motor plan and thought to precede sensory feedback–is greater than that of amputees. If true, this suggests that one-handers may not be able to refine their model enough to create an accurate template of their artificial arm. Another interpretation of this result would be that these individuals have a noisy action-selection process (Adams et al., 2020). However, since this deficit was dissociated from our key measure of absolute reaching error, we believe other mechanistic deficits might be at play.”

12. Line 473: missing 'and' between 'arm' and 'therefore'.

Thank you, this has been corrected.

13. Line 506-509: what is a "trial set"? and does this mean that there could be more than 6 practice trials, depending on the performance of the participants? If yes, this should be made more clear.

This has now been clarified as follows:

“For each arm, the task began with a set of 6 practice trials (using targets not included in the task target set), presented to the participant. Performance (accuracy) during these trials was not monitored, however, in cases where the participant did not fully follow the instructions (e.g., participant started moving before the beginning of the trial) or the participants requested more practice, the 6 practice trials were repeated again, following necessary adjustments.”

14. Line 514-515: It should be mentioned here how the trial end was determined during the experiment (it is mentioned later, in the data analysis section, saying 1 sec after movement initiation).

This has now been clarified as follows:

“In each trial, participants reached to a visual target (1.5 cm × 1.5 cm square) presented in one of 60 predefined locations (see Figure 1B). To encourage participants to perform fast-reaching movements, a maximum movement time of 1 sec per reach was imposed. At the end of a trial (1 sec following movement initiation), the target changed colour to blue to indicate the reach has ended and the endpoint position was recorded.”

Reviewer #2 (Recommendations for the authors):The only other suggestion that I have is that I do not believe the terms used by the authors to refer to their subject populations are appropriate or in keeping with best practices.I believe "One handers" is entirely inappropriate (and both groups only have one hand, so it is technically ambiguous).Best-practice is to refer to people and then the condition – so it would be persons with congenital deficiency, or persons with amputation, rather than "one handers" and "amputees".

We agree with the reviewer that this is not the best term to identify our participants, and that we should move away from it in future publications. After some discussions with representatives from this community, the preferred term is ‘individuals with a congenital (or acquired) limb difference’. We use this terminology when referring to individuals, and refer to the group as the ‘congenital group’ (in contrast to the amputees which we now refer to as the acquired group).

[Editors' note: further revisions were suggested prior to acceptance, as described below.]

The manuscript has been improved but there is one remaining issue that we would like to see addressed:Reviewer 1 asks to include the feedback correction analysis in the paper with clear statements about why initial errors may be a confound. This will allow the reader a complete picture of the analysis. To quote the reviewer:"The authors have successfully addressed almost all of my comments and concerns. The only issue that I'm a bit struggling with is the suggested mechanism that differences between the congenital and acquired groups is a problem with visually-based corrective movements, which has now become a central potential explanation in the paper. I acknowledge that I was the one proposing it, and I'm ok with suggesting it in the paper given the different results between the reaching and 2D localization task. However, in their response letter, the authors included a feedback correction analysis that suggests no difference between the groups, and it seems that they have decided not to include it in the manuscript. I agree that the results might be confounded by initial errors, but I wonder if this should also be included in the paper despite the lack of difference, specifying the potential reasons, and inviting future studies to examine it in a more controlled manner.In addition, the authors instructed participants to avoid feedback correction in the reaching task, and I think it will be helpful to explain how the interpretation fits with this instruction."

We are very happy to include the feedback correction analysis in the paper. Though, please note that it is impossible to infer from this analysis that there is no difference between the groups in feedback correction, as indicated by the inconclusive Bayes Factor. We believe inconclusive results should be interpreted with more nuance, and thus our main take-home message, as already highlighted in our previous revisions, is that future studies should look into the underlying mechanism for the increased reaching errors we document here.

Below are the changes and additions made to the manuscript.

Results:

“As a complementary approach, we also compared the angle of the corrective movement, which follows the initial movement. While participants were instructed to perform ballistic ‘one-shot’ reaching movements, this later phase of the movement is still known to be influenced by sensory feedback (Elliott et al., 2001), and thus may provide additional insight into the contributing factors to the absolute reaching error. For each trial, we’ve calculated the absolute angle between the direction vector at maximum speed and the direction vector at the end of the movement. A perfectly straight reach (with no corrective movements) would therefore have a corrective angle of 0. Using the same ANCOVA approach described above, we found inconclusive evidence for group differences in artificial arm corrective angles (*F_(_*_2,47)_=2.19, *p*=0.12, BF_Incl_=0.67). Within the congenital group, corrective angles were highly correlated with initial error angle (*r*_(16)_=0.76, *p*<0.001) and not correlated with motor variability (*r*_(16)_=0.35, *p*=0.16, BF_10_=0.73). In other words, individuals who made larger initial errors had more to correct for and therefore showed bigger corrective movements. However, individuals who made larger corrective movements weren’t necessarily more accurate in reaching the target. Given this inherent dependency, this measure does not provide us with an opportunity to decouple the contribution of the feedback correction from the size of the feedforward error.”

Discussion:

“As such, by elimination, our evidence suggests that the process of using visual information to perform corrective movements isn’t as efficient in the congenital group. We note that we were unable to capture this potential deficit when focusing solely on the corrective phase of the ballistic movement, likely due to the strong coupling with the initial movement phase. Nevertheless, this idea is consistent with previous evidence showing individuals with a congenital limb-difference have impaired processing of visual hand information (Maimon-Mor, Schone, et al., 2020). This interpretation is also compatible […] However, more work testing this suggested mechanism directly needs to be carried out to consolidate our interpretation.”

Methods:

“Corrective movements were defined as the absolute angle between the direction vector at peak velocity and the direction vector at the end of the movement.”